# A Schottky-Type Metal-Semiconductor-Metal Al_0.24_Ga_0.76_N UV Sensor Prepared by Using Selective Annealing

**DOI:** 10.3390/s21124243

**Published:** 2021-06-21

**Authors:** Byeong-Jun Park, Jeong-Hoon Seol, Sung-Ho Hahm

**Affiliations:** School of Electronic and Electrical Engineering, Kyungpook National University, Daegu 41566, Korea; qudwns27@knu.ac.kr (B.-J.P.); jeonghoonseol@knu.ac.kr (J.-H.S.)

**Keywords:** aluminum gallium nitride (AlGaN), UV, UV-to-visible rejection ratio (UVRR), local breakdown

## Abstract

Asymmetric metal-semiconductor-metal (MSM) aluminum gallium nitride (AlGaN) UV sensors with 24% Al were fabricated using a selective annealing technique that dramatically reduced the dark current density and improved the ohmic behavior and performance compared to a non-annealed sensor. Its dark current density at a bias of −2.0 V and UV-to-visible rejection ratio (UVRR) at a bias of −7.0 V were 8.5 × 10^−10^ A/cm^2^ and 672, respectively, which are significant improvements over a non-annealed sensor with a dark current density of 1.3 × 10^−7^ A/cm^2^ and UVRR of 84, respectively. The results of a transmission electron microscopy analysis demonstrate that the annealing process caused interdiffusion between the metal layers; the contact behavior between Ti/Al/Ni/Au and AlGaN changed from rectifying to ohmic behavior. The findings from an X-ray photoelectron spectroscopy analysis revealed that the O 1s binding energy peak intensity associated with Ga oxide, which causes current leakage from the AlGaN surface, decreased from around 846 to 598 counts/s after selective annealing.

## 1. Introduction

Nitride materials are attractive candidates for various optical applications due to their wide direct bandgaps and highly crystalline qualities following epitaxial growth [1,2,3]. In particular, nitride-based semiconductors have been broadly used in important applications such as high-power/high-speed electron devices, visible/UV laser diodes, LEDs, and UV sensors [4,5,6]. Ternary alloys, such as aluminum gallium nitride (AlGaN), are suitable materials for UV sensors due to the capability of detecting specific wavelengths based on the Al content, as well as their remarkable robustness toward harsh environments.

Recent studies have intensely explored device structures and experimental techniques to improve the performance of UV sensors [7,8,9,10,11,12]. Despite much progress, UV sensor technologies are still limited by high leakage current and poor UV-to-visible rejection ratio (UVRR) values, which are attributed to the low crystalline qualities of the ternary epitaxial layers and high defect densities. Rapid thermal annealing (RTA) is recognized as one of the methods to overcome these limitations because it improves device performance metrics that include trap density, device uniformity, and contact behavior [13,14,15,16,17,18,19,20].

Recently, a selective annealing method that employs dielectric breakdown changed the contact behavior of a nitride-based UV sensor, but the structural and electrical effects of the selective annealing process are not well understood [21]. Selective thermal annealing exploits Joule heating via a current that is generated following the local breakdown of the insulator in a metal-insulator-metal structure. Currently, specific electrodes can be selectively annealed only by using a breakdown voltage that cannot easily be scaled-up to larger areas without sophisticated fabrication techniques and equipment. These advantages have motivated further research into improving the reliability of selective annealing processes.

In this work, we investigated the effects of selective annealing on asymmetric MSM AlGaN/GaN UV sensors that were epitaxially grown on a sapphire substrate and have Al compositions of 24%. We used a Ti/Al/Ni/Au metal scheme in the sensors that minimized the degradation of device performance and exhibited reasonable ohmic behavior after selective annealing. The electrical and UV optical characteristics were analyzed both before and after selective annealing to enable a comparison. X-ray photoelectron spectroscopy (XPS) analysis provided insight into the leakage current on the AlGaN surface. Cross-sections of the selectively annealed regions were visualized by using transmission electron microscopy (TEM) and high-angle annular dark-field (HAADF) scanning transmission electron microscopy (STEM). Further analyses by STEM energy dispersive spectroscopy (EDS) elucidated the effects of selective annealing on the interfaces in the device.

## 2. Materials and Methods

Figure 1 shows cross-sectional and three-dimensional schematics of the fabricated asymmetric MSM AlGaN UV sensor. The AlGaN/GaN epitaxial layer and the GaN buffer layer were grown via metal-organic chemical vapor deposition on a (0001) sapphire substrate. The GaN layer was 3.5 µm thick and grown by using trimethylgallium (TMGa) and ammonia (NH_3_) sources, and the 300 nm bulk AlGaN layer was grown by using trimethylaluminum (TMAl) as an additional source. Wafers were first exposed to acetone, methanol, SPM (sulfuric acid (H_2_SO_4_): hydrogen peroxide (H_2_O_2_), 3:1), SC2 (hydrochloric acid (HCl): H_2_O, 1:1), and a buffered oxide etchant for 15, 5, 3, 3, and 1 min, respectively, to remove contaminants. After cleaning, the bottom electrodes were patterned by using photolithography. Next, Ni (30 nm) and Ti/Al (30/50 nm) were deposited by using electron-beam (e-beam) evaporation and lift-off processes. Subsequently, a 30-nm-thick insulating layer of aluminum oxide (Al_2_O_3_) was deposited by using atomic layer deposition (ALD) at a processing temperature of 300 °C. The insulator was selectively removed by dry etching with plasma to generate a contact hole. An ashing process was performed to remove the remaining organic matter and negative photoresist. Finally, after patterning the top electrodes and a dummy pad for selective annealing, an Ni/Au (30/50 nm) film was deposited by using e-beam evaporation and lift-off processes.

The area of the asymmetric MSM AlGaN UV sensor was 400 × 400 µm, the finger length was 200 µm, and both the finger width and the gap between the finger electrodes were 10 µm. The electrical characteristics of the sensors were measured by using an Agilent HP4156C parameter analyzer, while their spectral responsivity was characterized by using a 150 W Xenon arc lamp with a monochromator (Oriel 74,000) and a power meter (Newport 1930C).

For selective local annealing, the oxide was layered between metal electrodes 1 and 2, as shown in Figure 1a. A bias voltage sufficient to break the insulating layer between electrodes 1 and 2 enabled a high current that generates heat, thereby producing the annealing effect.

Table 1 reports the conditions for the selective annealing that were acquired using the Agilent HP4156C parameter analyzer. A bias voltage was applied to the Ti/Al/Ni/Au electrode for a total of 51 times from 39 to 40 V at intervals of 0.02 V to produce a continuous annealing effect. The maximum current was set to 50 mA.

## 3. Results

Figure 2 shows photoluminescence (PL) spectra obtained from the epitaxial wafer at room temperature before fabricating the sensor. High Al content increases the lattice mismatch of AlGaN and GaN, resulting in a poor-quality epitaxial layer: the AlGaN epitaxial layer showed a larger full-width at half-maximum (FWHM) compared to GaN. A photomicrograph of the fabricated asymmetric MSM AlGaN UV sensor is shown in Figure 3, where the contact edges were changed after the selective annealing. Table 2 summarizes the parameters for the MSM AlGaN UV sensor. The bandgap energy ($E_{g}$) as a function of Al composition can be obtained by applying the following equation [22]:(1)$$E_{g}=3.42+2.86x-x\left(1-x\right)\left[\mathrm{eV}\right]\;0<x<1$$

Figure 4 shows the electrical characteristics of the MSM AlGaN UV sensor before and after selective annealing. As shown in Figure 1b, all electrical characteristics of the asymmetric MSM AlGaN UV sensor were measured by applying a bias voltage to the Ni/Au electrode. For the MSM structure, the *I-V* characteristics were asymmetric due to the difference in the work functions of Ni (5.04 eV) and Ti (4.33 eV); the dark current density under a reverse bias was lower than that under a forward bias because Ni has a higher Schottky barrier height compared to that of Ti, as shown in the black curve of Figure 4a. At a forward bias of 2.0 V, the dark current density of the sensor prior to selective annealing was 7.6 × 10^−6^ A/cm^2^, but this dramatically increased to 0.8 A/cm^2^ after selective annealing, reflecting a 10^5^-fold increase. Figure 4 indicates that the Schottky behavior of the Ti/Al/Ni/Au electrode before selective annealing was changed to ohmic behavior after selective annealing. Dark current density at a bias of −2.0 V was 1.3 × 10^−7^ A/cm^2^ before annealing and 8.5 × 10^−10^ A/cm^2^ after annealing. Photo-response current shows a clear difference between the two; the highest photocurrent was observed under irradiation at 316 nm. Noisy currents are observed in the non-annealed device.

In Figure 5, analyses by XPS were conducted to interrogate Ga oxide that have imperfect structure and gain insights into the leakage current on the AlGaN surface. The sample was etched using an Ar ion gun of 500 eV, and the count/s for O 1s was measured every 5 s. The O 1s XPS spectra clearly show a peak near 531.6 eV that corresponds to Ga-O bond on the AlGaN surface, the peak intensity of which decreased from 846 to 598 counts/s after annealing, which indicates surface passivation [23,24,25,26].

Based on these results, we measured spectral responsivity under forward and reverse bias. Figure 6 shows the spectral responsivity of the MSM AlGaN UV sensors before and after selective annealing. The insets show its responsivity under forward bias. No spectral responsivity was observed except for 1 V before selective annealing because of the electron current supplied through the low barrier Ti/Al/Ni/Au electrode as shown in the inset of Figure 6a. After the annealing, the responsivity increased about 100 times overall and it disappeared at 1 V.

Under reverse bias, the non-annealed sensor exhibited a cut-off wavelength of 316 nm and a low UVRR due to a high leakage current. After selective annealing of the Ti/Al/Ni/Au electrode, a more reliable responsivity and high UVRR were obtained. Specifically, the UVRR at −7.0 V was 672, as shown in Figure 6b, which is 8 times higher than for the sensor before selective annealing; the UV and visible response currents for UVRR calculations were at 310 nm and at 400 nm.

To investigate the effect of selective annealing on the interfaces within the MSM AlGaN UV sensor, we visualized the cross-sections of the selectively annealed areas using TEM and STEM. As shown in Figure 1a, we identified two areas: one in which annealing was weak (area 1) and the other in which annealing was strong (area 2), based on the dielectric breakdown between electrode 1 (Ti/Al/Ni/Au) and electrode 2 (the dummy pad). Figure 7 shows TEM and high-angle annular dark-field (HAADF) STEM images of weakly annealed area 1, with Figure 7b–d depicting STEM images of the red box region in Figure 7a. Figure 7a,b show that each material layer was uniformly formed during the epitaxial growth process and e-beam evaporation. Figure 7c shows the presence of N inside the AlGaN layer as well as the Ti layer; nitrogen vacancies that were generated inside the AlGaN layer after N penetrated the Ti layer played an important role in altering the contact behavior. Ga was detected outside the sensor in the STEM analysis, as shown in the red box in Figure 7d, because a focused Ga ion beam was used. The STEM results in Figure 7d establish that a thin Ga layer exists between Al and Al_2_O_3_.

The elemental content in each layer was quantified across all layers of the MSM AlGaN UV sensor by using STEM EDS. Figure 8 shows the elemental content of Al, O, Ti, Ga, and N across the layers in weakly annealed area 1. Meanwhile, Table 3 includes the average elemental content in each layer, which establishes that the predominant element in each layer is as expected based on the controlled epitaxial growth and the results of the STEM analysis mentioned above. The plots in Figure 8 largely show sharp transitions in the elements across the various layers, thereby indicating the minimal diffusion of the elements from adjacent layers (except for N and O) that penetrated the Ti layer.

Figure 9 displays TEM and STEM images of strongly annealed area 2 in the MSM AlGaN UV sensor. Unlike weakly annealed area 1, the surface layer of area 2 (shown in Figure 9a) was uneven; the non-uniformity of the layers is more clearly shown in the STEM image in Figure 9b, which indicates that annealing caused interdiffusion of various metals. Specifically, it can be seen that the insulating layer of Al_2_O_3_ was much reduced as a result of dielectric breakdown, and Al and O apparently diffused into the adjacent layers. Moreover, Figure 9c shows that higher quantities of N penetrated the Ti layer in highly annealed area 2 versus weakly annealed area 1. Furthermore, a new layer formed between the Ti and Al layers (Figure 9b) that was established as predominantly Ga, as can be observed in the STEM image in Figure 9d.

Figure 10 shows the elemental content in the layers of strongly annealed area 2, while Table 4 summarizes the average elemental content in each layer. Compared to area 1 in the MSM AlGaN UV sensor, the EDS data for area 2 show rough boundaries between each layer. The diffusion of Al during annealing formed two peaks and blurred the boundary between the layers, the O content increased in most layers except for the Al_2_O_3_ layer, Ti slightly migrated into the Al layer, and the N content in the Ti layer increased significantly compared to weakly annealed area 1. Furthermore, a Ga layer that formed between the Ti and Al layers due to the strong annealing effect was clearly observable as Al and Ga peaks in Figure 10 and Table 4.

## 4. Discussion

Dark current density and UVRR are important factors in determining the performance of UV sensors. AlGaN UV sensors can measure both UV-A and UV-B depending on the bias voltage and are applicable in various applications using dual-wavelength detection, further raising the importance of the reduction in dark current density to improve sensor performance. In general, higher Al content causes lattice mismatches that lead to more defects, further resulting in poor epitaxial quality and high leakage current. Thus, we analyzed the selective annealing effects on AlGaN UV sensors with the lowest Al content among our samples.

The electrical characteristics of the AlGaN UV sensor with 24% Al before and after selective annealing were investigated. As shown in Figure 4a,b, under forward bias, the selective annealing process changes the rectifying contact of Ti/Al/Ni/Au electrode to near ohmic contact, which we suspect originated from the formation of nitrogen vacancies. During annealing, N easily reacts with Ti to form TiN, and the remaining nitrogen vacancies serve as donors on the AlGaN substrate and promote high electron concentration. This changes the contact behavior by reducing the Schottky barrier in the Ti/Al/Ni/Au electrode.

Under reverse bias, non-annealed sensors show high leakage current values that are partially due to poor interface quality and bulk defects. This originates from current leakage at the AlGaN surface via Ga oxides. In the atmosphere, Ga reacts with oxygen on the AlGaN surface to form thin Ga oxide layers such as Ga_2_O, GaO_2_, and Ga_2_O_3_. Among them, chemical bonds such as GaO_2_ act as a trap on the surface that leads to operation instability and performance degradation. In Figure 5, O 1s XPS spectral analysis at AlGaN surface is conducted and displayed. To obtain XPS results measured on the AlGaN surface, the sample was etched for 155 s (35 nm) using an Ar ion gun of 500 eV. The main peak of the O 1s spectra is centered at a binding energy of 531.6 eV, which corresponds to the Ga-O bond. After selective annealing, the O 1s peak intensity for Ga-O bond with a binding energy of 531.6 eV decreased, which means that Ga oxide on the AlGaN surface was reduced. This suggests a reduction in the oxide related traps that cause leakage current that leads to dark current density.

Moreover, the reduction in dark current density could have arisen from a decreased number of traps at the Ni/AlGaN interface. Although the electrodes were isolated, electron microscopy established that dielectric breakdown during annealing slightly impacted not only area 1 and area 2 but also a wide range of interdigitated fingers and surrounding insulators. We suspect that annealing affected the traps at the Ni/AlGaN interface because, under equilibrium conditions, the majority of them below the Fermi level are filled with electrons. Thus, the application of a reverse bias at the Ni/Au electrode would cause electrons captured by the interfacial traps to be emitted by trap-assisted tunneling, thereby contributing to the leakage current. Selective annealing is thought to passivate and reduce interfacial traps, thus resulting in a lower dark current density.

As shown in Figure 4b, the photocurrent density of the MSM AlGaN UV sensor was highest when irradiated with light at 316 nm, which corresponds to the E_g_ of AlGaN. Irradiation with light at a wavelength of 365 nm, which corresponds to the E_g_ of GaN, generates photoelectrons in the GaN layer. Some of these are blocked by the AlGaN barrier, which makes it impossible for them to contribute to the current flow. By comparison, for irradiation at 316 nm, photoelectrons generated in the AlGaN layer directly contribute to the current. Selective annealing reduced the dark current density and resulted in more reliable photoresponsive behavior.

In Figure 6, the Schottky barrier height between Ti/Al/Ni/Au and AlGaN was further reduced after selective annealing, eliminating the characteristics shown under 1 V and increasing the responsivity. This means that the contact behavior switched from rectifying to ohmic. Under reverse bias, Figure 6 also shows that selective annealing improves the UVRR by the reduced traps at the Ni/AlGaN interface. Specifically, the UVRR after selective annealing was 672 at −7.0 V, which was 8 times higher than in the non-annealed sensor. These results establish that selective annealing caused significant performance improvements, which potentially indicates that more reliable spectral responsivity and a higher UVRR could be achieved by optimizing the AlGaN layer quality as well as the annealing conditions.

The impact of selective annealing on the interfaces and surfaces inside the sensor was interrogated by using TEM, which revealed two key insights. First, as shown in Figure 7 and Figure 9, the higher annealing strengths resulted in greater penetration of elements into the surrounding layers. For example, the N content that penetrated the Ti layer was higher in strongly annealed area 2 than in weakly annealed area 1. This explains the shift from Schottky behavior in the non-annealed Ti/Al/Ni/Au electrode to ohmic behavior in the annealed one. Second, Figure 9 and Figure 10 show that Ga inside the AlGaN layer migrated to the Ti layer and formed a new layer, the effect of which on the change in the contact behavior of the device requires further study. In either case, these results establish that the selective annealing process provides enough energy to break the Ga-N bonds and produce nitrogen vacancies that can influence the semiconductor properties, which changes the device performance. In addition, as reported in Table 4, the O content after selective annealing increased in all layers except the insulating Al_2_O_3_ layer. This could indicate that atmospheric O penetrated the device during selective annealing or that O in the device diffused into the layers, or a combination of both. We anticipate that performing the annealing process, which involves dielectric breakdown, in an N_2_ atmosphere will reduce the O content.

Unlike the conventional RTA processes, the proposed method requires simple process at the later stage of fabrication process without costly equipment. It is possible to develop a high-thruput annealing system using a probe card designed for heat treatment only for a specified area on the wafer so that there could be no thermal damage overall wafer. They may require minimal dummy pads to reduce the material waste [27].

## 5. Conclusions

Asymmetric MSM AlGaN UV sensors with 24% Al were fabricated to investigate the effects of selective annealing on device performance. A Ti/Al/Ni/Au metal scheme was employed for the electrode to obtain reliable device performance after dielectric breakdown. Under a forward bias of 2.0 V, the selective annealing substantially increased the dark current density from 7.6 × 10^−6^ A/cm^2^ to 0.8 A/cm^2^, which was attributed to nitrogen vacancies generated by the selective annealing that altered the contact behavior of the Ti/Al/Ni/Au electrode. Under reverse bias, the dark current density at a bias of −2.0 V was 8.5 × 10^−10^ A/cm^2^ and the UVRR at a bias of −7.0 V was 672. The results of the XPS analysis of the sensors show that annealing reduced the peak intensity of the O 1s binding energy associated with Ga oxide on the AlGaN surface from around 846 to 598 counts/s. These results demonstrate a remarkable performance improvement as a result of the selective annealing, which probably arose from surface passivation and a reduction in the number of traps at the metal/AlGaN interface.

These results suggest that selective annealing by dielectric breakdown is a facile technique involving localized heating that does not require additional fabrication processing steps or equipment and shows it is useful for improving the performance of sensor devices.

## Figures and Tables

**Figure 1 sensors-21-04243-f001:**
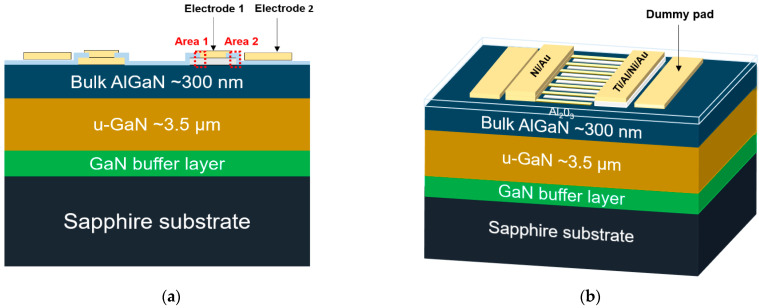
Schematics of the asymmetric MSM AlGaN UV sensor with 24% Al: (**a**) a cross-sectional view and (**b**) a three-dimensional view.

**Figure 2 sensors-21-04243-f002:**
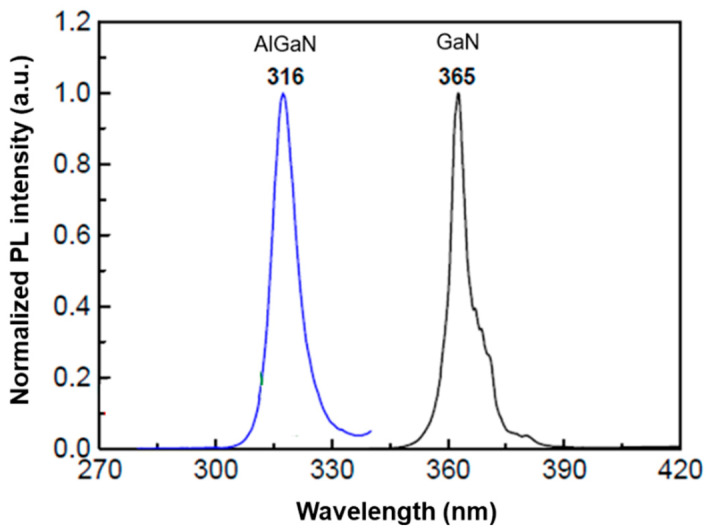
PL spectra of the epitaxial wafer before fabrication of the asymmetric MSM AlGaN UV sensor at room temperature.

**Figure 3 sensors-21-04243-f003:**
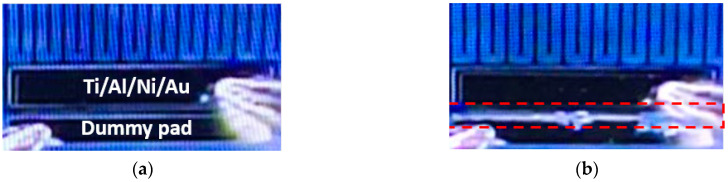
Images magnified 90× using a photomicroscope of MS TECH’s MST-8000C probe station: top views of an asymmetric MSM AlGaN UV sensor (**a**) before and (**b**) after selective annealing (annealed region is marked with dotted box).

**Figure 4 sensors-21-04243-f004:**
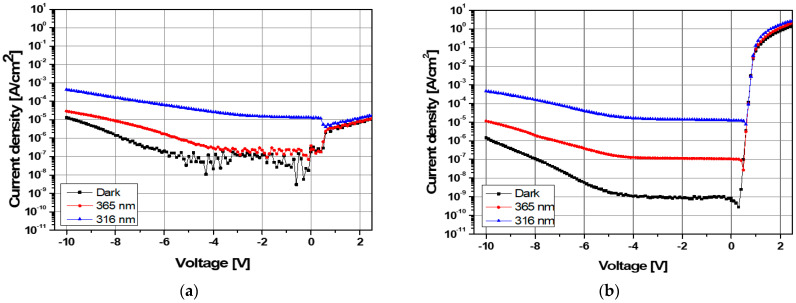
Dark state and UV photoresponsive *I-V* of the asymmetric MSM AlGaN UV sensor (**a**) before and (**b**) after selective annealing.

**Figure 5 sensors-21-04243-f005:**
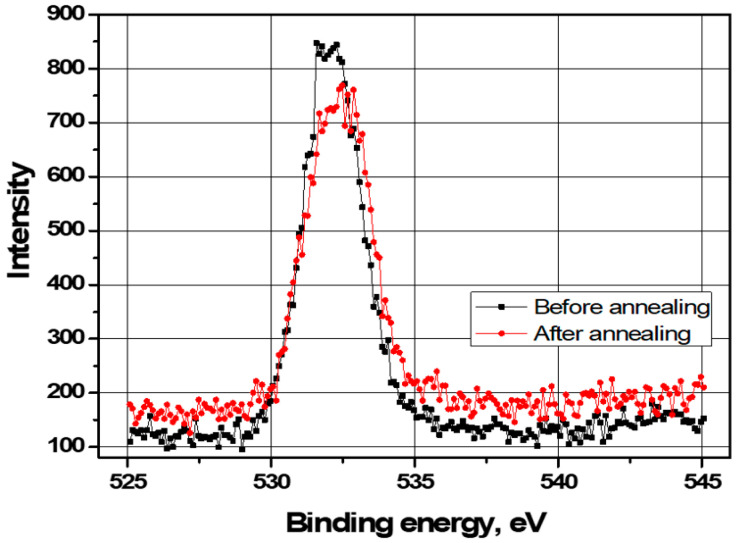
O 1s XPS spectra associated with Ga oxide on the AlGaN surface of the MSM AlGaN UV sensor before and after selective annealing.

**Figure 6 sensors-21-04243-f006:**
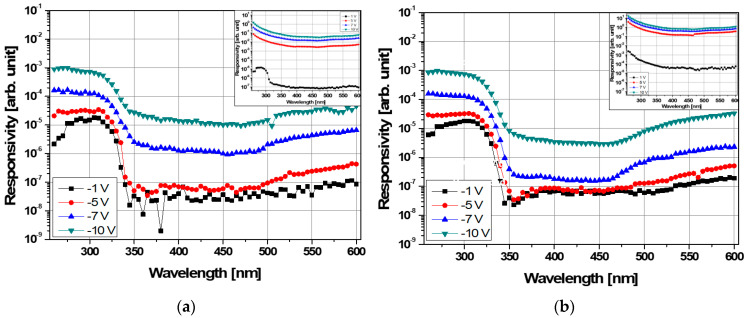
Spectral responsivity of the MSM AlGaN UV sensor (**a**) before and (**b**) after selective annealing under reverse bias. The insets show the spectral responsivity under forward bias.

**Figure 7 sensors-21-04243-f007:**
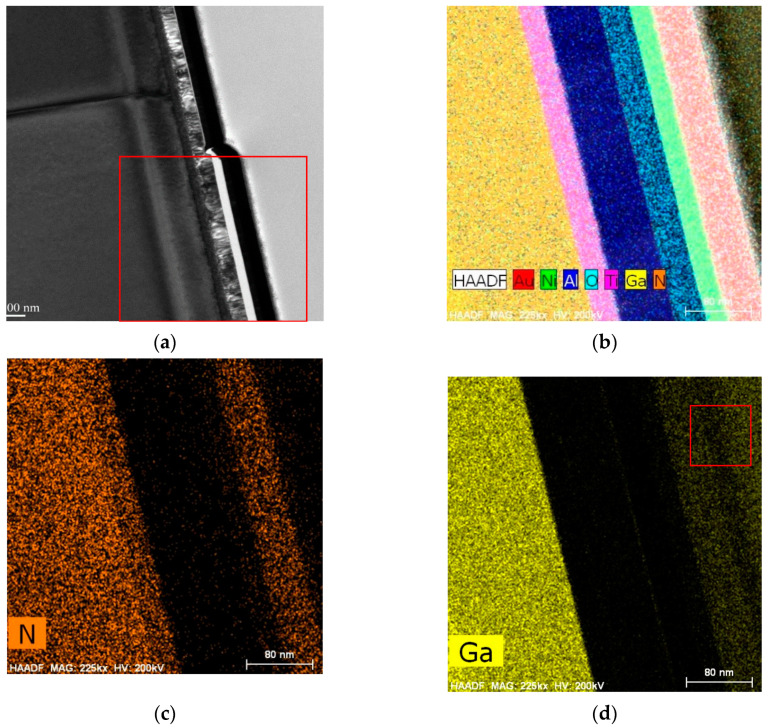
(**a**) TEM and (**b**–**d**) HAADF STEM images of weakly annealed area 1 in the MSM AlGaN UV sensor.

**Figure 8 sensors-21-04243-f008:**
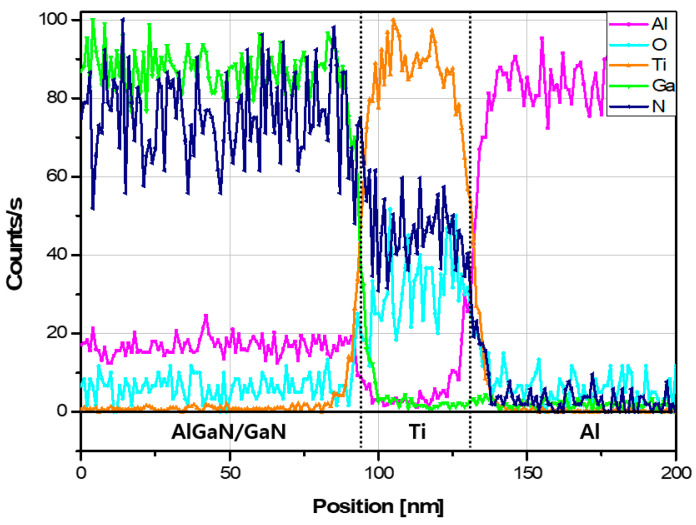
Elemental content in each position in weakly annealed area 1.

**Figure 9 sensors-21-04243-f009:**
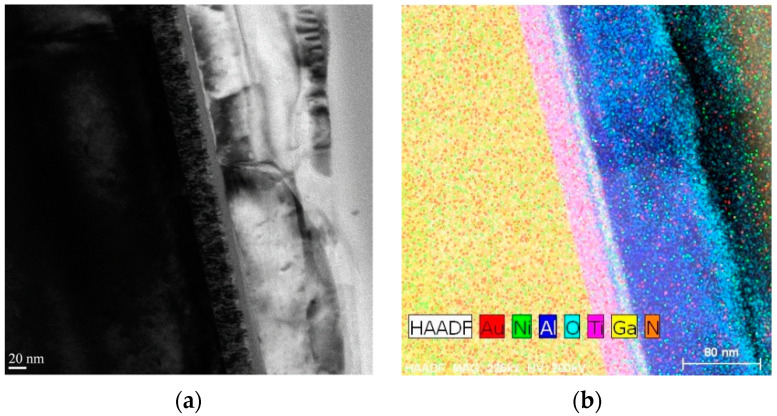
(**a**) TEM and (**b**–**d**) HAADF STEM images of strongly annealed area 2 in the MSM AlGaN UV sensor.

**Figure 10 sensors-21-04243-f010:**
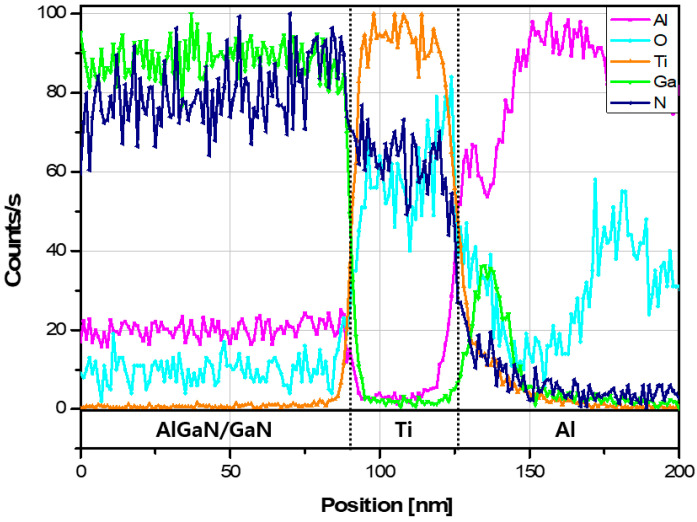
Elemental content in each position in strongly annealed area 2.

**Table 1 sensors-21-04243-t001:** Selective annealing conditions for dielectric breakdown.

Parameter	Value
Mode	Sweep
Start (V)	39
Stop (V)	40
Point	51
Step Size (mV)	20
Compliance (mA)	50

**Table 2 sensors-21-04243-t002:** Important parameters for the MSM AlGaN UV sensor.

Parameter	Value
Al content (%)	24
PL peak (nm)	316
FWHM (nm)	6
$E_{g}$ (eV)	3.92

**Table 3 sensors-21-04243-t003:** The average content of the elements within each layer in area 1.

Element	Number of Counting (Arbitrary Unit)				
	AlGaN	Ti	Ti-Al Interface	Al	Al_2_O_3_
Al	17.0	4.9	36.3	76.7	35.9
O	6.4	31.0	24.4	12.4	65.9
Ti	1.7	90.9	16.4	4.2	0.3
Ga	87.8	8.7	2.6	2.4	0.9
N	75.1	49.8	35.9	4.9	4.3

**Table 4 sensors-21-04243-t004:** The average content of the elements within each layer in area 2.

Element	Number of Counting (Arbitrary Unit)				
	AlGaN	Ti	Ti-Al Interface	Al	Al_2_O_3_
Al	20.3	4.1	68.1	78.4	44.1
O	10.4	55.5	28.6	33.8	64.7
Ti	1.6	84.7	20.2	6.1	0.1
Ga	88.6	3.9	25.5	5.6	1.9
N	78.7	68.2	14.9	7.1	2.6

## Data Availability

Not applicable.

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
