# Peer review of "A Schottky-Type Metal-Semiconductor-Metal Al0.24Ga0.76N UV Sensor Prepared by Using Selective Annealing"

_sensors, 2021, doi:10.3390/s21124243_

Round 1

Reviewer 1 Report

The paper titled A Schottky-Type Metal-Semiconductor-Metal Al0.24Ga0.76N UV Sensor Prepared by Using Selective Annealing deals with UV detectors annealed by selective annealing caused by voltage applying. The paper is very interesting for the scientific community especially interested in UV photodetector fabrication. However, the presented version of the paper need to be revised. Bellow I listed my detailed remarks:

  • what authors think about classical annealing, I mean RTA process? Did you perform some comparative study of selective annealing detectors with RTA processed UV sensors?
  • line 34 - UVRR  -in my opinion this acronym mean UV-to-visible rectification ratio instead rejection  ratio
  • line 34 - authors have to specify UV and VIS wavelength used for UVRR calculation
  • line 64: NH3 instead of NH3
  • line 96: authors presented PL spectrum of MSM AlGaN UV detector, is it mean that this PL was measured on structure with metal contact?
  • line 112 - work function have to be mentioned with units, i.e. eV
  • Fig. 3b - why authors do not shown similar curve for both detectors ( as grown and annealed). Moreover the Fig. 3 description is not correct because Fig. 3b shown result only for one type of detectors
  • authors provide XPS results and conclude that annealing reduce the amount of Ga oxide. What kind of oxide Ga2O3 or Ga2O?
  • lines 138-140 -  During forward bias, no spectral responsivity was observed because the electrons supplied from the Ti/Al/Ni/Au electrode easily passed the lower Schottky barrier of  the Ti/Al/Ni/Au contact.  - please include example spectra in Figure 5
  • XPS description in conclusion is more detailed than in main text

Reviewer 2 Report

This paper reports on the fabrication of MSM AlGaN UV sensors using a selective annealing technique, which uses the local heating generated by the breakdown of a MIS system placed close to the detector active area.
In particular, the electrical characteristics of the sensor, as well as their optical response, could be explained on the base of TEM analyses, which showed an interdiffusion of the metal layers (associated to a transition to Ohmic contact).

The paper is surely original and fully in line with the scope of the journal. However, I recommend to clarify some points before publication:

1) Coparative results obtained by conventional RTA annealing on the Ti/Al/Ni/Au electrode should be presented to strenghten the discussion. As an example, one could fabricate an annealed (800°C)  Ti/Al/Ni/Au ohmimc electrode first, and then add the second interdigit Ni/Au electrode. Such a process would be more controllable and uniform. It would be very interesting to have such a comparison, to judge how controllable is the proposed method.

2) The surface morphology of the pads and the geometry of the interdigit fingers (edge acuity) after this local annealing process should be shown (e.g. AFM and SEM top views could be useful).

3) The possibility of apply this method on a large scale is questionable (as the method seems to be difficult to control and to be reproducible). In fact, the reviewer understands that the properties of the metal layer after annealing are not uniform, as different regions are observed . Moreover, the use of the dummy pads can represent a significant waste of material (in this way you can fabricate less devices inside a wafer), thus diminuishing the advantage of using such a method. Please comment this aspect in the text.

Round 2

Reviewer 1 Report

I appreciate the revision made by authors. In my opinion paper can be published in Sensors.